# Ecological Virtuous Selves: Towards a Non-Anthropocentric Environmental Virtue Ethic?

**Damien Delorme** [1,*] **, Noemi Calidori** [2] **and Giovanni Frigo** [3]

1 Faculty of Geosciences and Environment (FGSE), Institute of Geography and Sustainability (IGD), University of Lausanne, 1015 Lausanne, Switzerland
2 Department of Law, Roma Tre University, 00154 Rome, Italy; ncalidori@uniroma3.it
3 Institute for Technology Assessment and Systems Analysis (ITAS), Karlsruhe Institute of Technology (KIT), 76131 Karlsruhe, Germany; giovanni.frigo@kit.edu
* Correspondence: damien.delorme@unil.ch

**Abstract:** Existing predominant approaches within virtue ethics (VE) assume humans as the typical agent and virtues as dispositions that pertain primarily to human–human interpersonal relationships. Similarly, the main accounts in the more specific area of environmental virtue ethics (EVE) tend to support weak anthropocentric positions, in which virtues are understood as excellent dispositions of human agents. In addition, however, several EVE authors have also considered virtues that benefit non-human beings and entities (e.g., environmental or ecological virtues). The latter correspond to excellent character dispositions that would extend moral consideration and care for the benefit of non-human beings, entities, or entire ecosystems. In this direction, a few authors have argued that EVE could be considered non-anthropocentric insofar as it could: (a) promote non-human ends, well-being, and the flourishing of non-human beings and entities; (b) involve significant relations to non-humans. Drawing from different traditions, including ecofeminism and care ethics, we argue for a broader notion of self and a decentered notion of virtues. The broader notion of selfhood corresponds to the "ecological self", one that can be enacted by both human and non-human beings, is embedded in a network of relations, and recognizes the more-than-human world as fundamental and yet indispensable otherness. We suggest that this broader notion of agency allows for an expansive understanding of virtues that includes a-moral functional ecological virtues, which can be exercised not only by humans but also by certain non-human beings. This alternative understanding of selfhood and ecological virtues within EVE could have several theoretical and practical implications, some of which may enable different types of agencies and transform collective action.

**Keywords:** ecological self; ecofeminism; care ethics; environmental virtue ethics; non-anthropocentrism; ecological virtues; agency

## 1. Introduction

A cautious but philosophically reasonable answer to the leading question of this collection—"Is Environmental Virtue Ethics a Virtuous Anthropocentrism?"—might be "It depends". Of course, it depends on the specific meaning of some of these loaded terms (e.g., "virtue" or "anthropocentrism"). However, the response is also based on some underlying theoretical assumptions, such as the type of agent or actor that is considered capable of demonstrating virtues, and regarding who or what may potentially benefit from certain virtues. In this paper, we question some assumptions regarding the notion of agency and virtue elaborated so far in environmental virtue ethics (EVE) and, as an alternative approach to EVE, we explore non-anthropocentric notions of subjectivity and agency that are not yet moralized. More specifically, we propose that it is possible to answer "Not necessarily" to the question above, provided that the notions of agency and virtues are decentered from human subjectivity, that is, when they become *de*-anthropocentrized. Our aim is to explore the metaphysical, ontological, and ethical conditions for including

non-anthropocentric perspectives within EVE. Our research question is the following: Can a broader notion of self combined with an extended notion of virtues constitute the preconditions for developing a non-anthropocentric approach in environmental virtue ethics? Alternatively, one may ask: can non-anthropocentric notions of agency and virtue become the basis for virtuous non-anthropocentrism in EVE?

Most EVE accounts that have emerged over the past twenty years tend to be weakly anthropocentric. Although this may depend on several reasons, here we suggest that it hinges on assumptions regarding both the notion of agency and that of virtue. First, even though most EVE scholars acknowledge that the human exercise of some virtues may benefit non-human[1] beings and entities (i.e., environmental or ecological virtues), they do not go so far as to claim that *non-human agents* can exercise virtues. This may depend on the metaphysical assumption that only humans are *normative* agents or on the fact that there is a long-standing tradition of thinking primarily about *human* virtues. Second, most accounts in EVE may have consistently assumed a narrow notion of virtue as *moral* virtue, which is a normative human endeavor or an exclusively human disposition or practice. Let us now consider existing alternatives to these views.

*Agency*. Although, within EVE scholarship, there seems to be no account that explicitly proposes that non-human agents can exercise virtues, some authors in the broader field of environmental ethics have already suggested extended notions of agency and selfhood (e.g., ecological self) that could be considered also in the context of EVE. Thinkers related to ecofeminist and care ethic traditions have often suggested more expansive and relational notions of selfhood and agency. Val Plumwood, for example, writes that "the ecological self recognises the earth other as a centre of agency or intentionality having its origin and place like mine in the community of the earth, but as a different centre of agency, which limits mine." [1] (p. 159). Thus, it is possible to conceive of agency beyond human agency.

*Virtue*. In EVE literature, "environmental or ecological virtues"[2] have been primarily conceptualized as types of excellent dispositions or behaviors that are exercised by human agents and benefit non-human beings and entities. However, there are at least two possible alternatives to the above-mentioned majority view in EVE. On the one hand, since Aristotle proposed that virtue can also be a quality of non-human beings (e.g., horse) or entities (e.g., knife), it seems possible to conceive of virtues in moral, a-moral, or functional ways. In this sense, a sharp knife is *functionally* virtuous insofar as it cuts the paper well, or the horse is *functionally* virtuous because of being courageous in (human) battle. In this view, however, the knife and the horse are functionally virtuous while only the human person is potentially capable of being morally virtuous. On the other hand, an alternative option could come from cultural/natural anthropology. Drawing upon Descola's anthropology of nature [3,4] and what we might call "non-modern (or non-naturalistic) cultures" (i.e., animism, analogism and totemism), certain virtues may not need to be exclusive to certain agents but could be shared among humans and non-humans within a specific context.

Building on these suggestions, we argue that broadening the notion of self and decentering that of virtues—both in non-anthropocentric terms—would represent the preconditions for developing accounts of non-anthropocentric EVE. Although we do not believe that this is necessarily a fruitful path forward, we suggest that it is important to consider how promising and doable such accounts might be [5]. At the core of the paper, we argue that the notion of self can broaden to become an "ecological self", while that of virtue can expand to include functional ecological a-moral virtues that can also be exercised by non-human agents. Envisioning virtuous agency in non-anthropocentric terms allows for a likewise non-anthropocentric broadening of "ecological virtues", which would represent types of excellent dispositions that can be exercised also by non-human agents. Of course, this is possible because the notion of self is conceptualized as an ecological self. This focus on the ecological self is in line with what Callicott writes when affirming that "the nature of the self—or better how to conceive of and to experience the self—is the central philosophical question of environmental ethics and indeed of ecophilosophy" [6] (p. 11). In this sense, both agency and virtue are decentered or de-anthropocentrized. These theoretical alternative assumptions or proposed theoretical

changes may represent the basis for developing consistent non-anthropocentric accounts in EVE. Given the previous discussion, it should be clear that our thesis challenges the notion of agency in VE (and EVE), as well as the notion of ecological virtues elaborated so far in EVE.

In practice, we maintain that the ecological self can be either a human person or a non-human being and that, while the former can exercise both moral and a-moral virtues, certain species within the latter may only be capable of exercising a-moral virtues. The two reconceptualized notions of "ecological self" and "ecological virtue" would antagonize the separatist function of the so far hegemonic anthropocentric EVE discourse and foster a theoretical and practical precondition for reconnecting human and non-human entities and beings. In this sense, we present an "extensionist" strategy, a theoretical proposal that decenters and broadens both the notion of selfhood and that of virtue, allowing for a compositionist (or non-separatist) framing. This may serve as the ground on which to build non-anthropocentric accounts of EVE that imagine and devise environmental and climate policies differently.

Section 2 illustrates four different ways in which both human and non-human agents could exercise virtues that may benefit either human or non-human subjects or ends. Section 3 describes how it is possible to broaden the notion of self to ecological self. Section 4 develops our proposal to consider functional ecological virtue as a way to transform EVE in a non-anthropocentric perspective and considers some objections. In conclusion, we highlight the benefits of thinking critically about agency and ecological virtues for (human) ethics generally, and we discuss some implications of our proposal for EVE specifically.

## 2. Variations of Agency Regarding Virtue

Virtue theory has long emphasized the polysemy of the concept of virtue. Leaving aside ulterior types of virtues, such as intellectual or epistemic virtues, here we focus on virtues within the virtue ethics tradition. Reflecting on a possible core concept to establish a coherent virtue ethics, Alasdair MacIntyre distinguished "three very different conceptions of a virtue [...]: a virtue is a quality which enables an individual to discharge his or her social role (Homer); a virtue is a quality which enables an individual to move towards the achievement of the specifically human telos, whether natural or supernatural (Aristotle, the New Testament, and Aquinas); a virtue is a quality which has utility in achieving earthly and heavenly success (Franklin)" [7] (p. 122). MacIntyre then notes that each conception refers to a pre-requisite conception of what constitutes a practice, the telos of a human life, and a moral tradition.

This diagnosis seems to reveal an axiom of virtue ethics; virtues are not only exercised by humans, but they express the superior powers of human normativity, which also distinguish them from non-humans and insist on their sovereignty via their practical wisdom. We could call this axiom, in a nod to Routley, the BHC (basic human chauvinism) of VE.

Can we contest this axiom? If we neutralize the anthropocentric assumptions—that practice, telos of a human life, and moral traditions, not only concern specifically human beings but also characterize their metaphysical supremacy—what would be a potentially core conception of (non-anthropocentric) virtue? Different strategies can be found in VE's distinctions.

We first consider the distinction between moral/non-moral virtues [8,9]. This distinction is usually used to question the supposed inherent relation between virtue and morality, leading, for instance, to consider contra-moral virtues [8] and not, as far as we know, to potentially expand the attribution of virtues to non-human agents.

We can also consider the distinction between eudaimonistic (Aristotelian tradition) and intuitionist accounts (Humean tradition) [10,11]. According to Huang, "the former explains virtue as *the character traits that contribute to human flourishing*, while the latter describes it as the *character traits that are simply admirable*" [12]. However, once again, this distinction remains within the prejudice that only humans could be virtuous. For instance, exploring a Daoist perspective on virtue ethics and following Zhuang Zhe, Huang only defined human

virtues; whereas, in our understanding, the principle Daoist virtue, namely a differentialist virtue: "respect diverse ways of life" [12], could endorse a non-anthropocentric meaning.

Another relevant strategy to consider the attribution of virtue to non-human traits of characters and actions could be suggested by the pluralistic virtue ethics developed by Christine Swanton [13]. She adopted a broad definition of virtue as "a good quality of character, more specifically a disposition to respond to, or acknowledge, items within its field or fields in an excellent or good enough way" [13] (p. 19). However, this disposition *to respond well to the demands of the world* is, implicitly and as far as we know, restricted to human agents, even though Swanton explained that her pluralistic view of virtue "avoids the problem of anthropocentrism" [13] (p. 50). In this sense, Swanton suggested that virtues might not be necessarily anthropocentric, and yet, she does not go as far as to propose that non-human beings can act virtuously (i.e., she does not challenge the exclusivity of human agency in VE).

Could a-moral virtue or functionalist virtue and the intuitionist account (recognizing excellences of different kinds) be extended to non-human agents as excellent dispositions capable of responding well to the world? Importantly, Aristotle already pointed out the linkages and differences between "*areté*" and "*ethike areté*"[3]. While the former is defined as "a perfect adaptation" [15] (p. 46) and can be applied to non-human beings, the latter denotes moral virtues that pertain to human morality. Interestingly, the BHC perspective assumes that *ethike areté* is superior to *areté*, but could we avoid a supremacist attitude and envision the exercise of virtuous dispositions as something that humans may share with other species? To illustrate how such an extension might be possible, we list the following four main ways to specify agency and the subject(s) affected by its exercise of both moral and a-moral virtues.

(a)  *Human agency affecting human(s)*. As anticipated above, this corresponds to the more traditional VE but is present also in EVE. To better understand it, consider the example of the human eye as presented by Aristotle. Such conception of virtue is typically weakly anthropocentric and is displayed in character traits such as compassion, attentiveness, attention, care, justice, etc., toward humans. According to Naess, embracing this type of moral orientation (i.e., "protecting Nature is protecting ourselves") could constitute a motivation for pragmatic ecological ethics.

(b)  *Non-human agency affecting human(s)*. In this case, the agent is a non-human being that is capable of exercising a virtue that has an effect on humans. It is interesting to point out that such a virtue would be described as an a-moral virtue when considering the non-human agent but potentially also as a moral virtue when considered from the point of view of the beneficiary (i.e., human(s)). For example, Aristotle described this version of virtuous behavior through the case of the brave horse, and it can be easily expanded to other instrumental relationships between animals and humans (e.g., animal labor/working force, food production, care labor). More broadly, this variation can be found in a lot of so-called "ecological services", or "nature's contribution to people" [16], or examples of symbiotic processes that benefit humans (for instance, see Margulis [17], Haraway [18]): breathing, digesting, pollination, filtration, providing food and shelter, and so forth. Accordingly, one might say that non-human agents that affect humans characterize every non-human precondition for human subsistence and flourishing. Often, these ecological capacities are turned toward human ends or made more efficient and productive through technical and technological means (e.g., devices, systems, processes). It goes without saying that many of such relationships are ambivalent and, like the Greek *pharmakon*, can designate either a poison or a medicine/cure (e.g., auto-immune diseases).

(c)  *Human agency affecting non-human(s)* (i.e., beings and entities). This option corresponds to EVE's distinctive contribution through the notion of "ecological virtues". Often, the effect(s) of such virtue(s) run the risk of being anthropomorphic in the sense that they may favor a human-centered conception of the "good/bad ends", excluding more pluralistic conceptions (i.e., from the point of view of the non-human(s) affected). This

approach comprises virtues similar to (a) but that affect non-human(s). A further risk of prioritizing non-human ends that has been highlighted several times in environmental ethics literature is that it could favor misanthropic, too radical, or eco-fascist conducts (e.g., some forms of radical environmentalism or Foreman's rewilding proposal). Although the potential sacrifices by humans and even of humans could cohere with the recognition of the intrinsic value of non-human beings and entities, this remains extremely controversial. Later, we suggest that human selves as ecological selves can act virtuously in favor of ecological ends, the ecological worth of which could be studied scientifically.

(d) *Non-human agency affecting non-human(s)* (i.e., beings and entities). In line with our thesis, it is possible to consider non-human beings as agents of a-moral functional ecological virtues that affect non-human beings, entities, or even ecosystems. Here, the distinctive and perhaps original element resides in the fact that the notion of self is broadened to include non-humans as potential agents (like in [b]), and the notion of virtue is decentralized as ecological virtue (like in [c]). In other words, non-human beings can act as ecological selves and are therefore considered agents capable of virtuous actions and behaviors that affect non-humans[4]. Similarly to [c], non-human beings can act virtuously by exercising a-moral functional ecological virtues and, similarly to [c], can affect positively different ecological dimensions and ends. As mentioned in [c], the "goodness" of these effects is a-moral and could be studied scientifically.

In the next two sections, we delve into the expanded notion of self as ecological self (Section 3) and that of virtue as non-anthropocentric ecological virtue (Section 4).

## 3. Broadening the Notion of Agency as Ecological Self

### 3.1. Three Traditional Conceptions of "Ecological Self" in Environmental Philosophy

At a basic level, adopting an ecological self means moving beyond a detached notion of selfhood towards one that acknowledges the fundamental importance of relationships among different species and to the ecosystems they live in. Put in the words of Karen J. Warren, in the *Encyclopedia of Environmental Ethics and Philosophy*, recognizing the ecological self means recognizing that "the self is not an isolated, immaterial Cartesian ego, soul or psyche in a physical body (lampooned as "the ghost in the machine"); rather, it is constituted by its relationships with others—just as in ecology the characteristics of various species are constituted by their relationships with other species and the abiotic environment" [19] (p. 231). From at least the late 1980s onwards, different versions of the ecological self have been developed. These have challenged the prevailing notion of selfhood found in European and North American philosophy, which hinges on atomistic metaphysical assumptions [20]. Instead, these alternative proposals are grounded on metaphysical assumptions that recognize the self—or perhaps better, a multitude of selves—as necessarily relational. In this view, humans are part of nature and deeply interconnected with natural beings, entities, and processes.

Despite these general and shared premises about the ecological self, here we discuss three main versions that have been elaborated so far. First, Arne Naess introduced the concept of the ecological self in a seminal article of 1987 entitled "Self-realisation: An ecological approach to Being"[5]. Among others, he drew from the *Gestalt* theory, Spinoza, and Eastern spiritual wisdom to offer a conceptualization of the ecological self based on the "process of identification" with others [22] (p. 35). He explained this process through an example doomed to become famous: once, Naess was looking through a microscope and a flea landed in the acid chemicals he was observing. Within a few minutes, the flea died, and Naess witnessed closely the flea's torturous attempts to live. He explains in these terms what he felt:

> "what I felt was naturally, a painful compassion and empathy. But the empathy was not basic, it was the process of identification, that 'I see myself in the flea'. If I was alienated from the flea, not seeing intuitively anything even resembling

myself, the death struggle would have left me indifferent. So there must be identification in order for there to be compassion [. . .]" [21] (p. 36).

Naess proposed that the concept of the self extends beyond the traditional understanding of it as simply an "ego" or a "social self". For ecological relationships to become part of our internal relationships, the process/phenomenon of identification is needed. As a result, for Naess, the pursuit of self-realization naturally leads human beings to take a greater interest in and concern for environmental issues. Thus, defending nature is equivalent to defending one-(ecological)-self.

A second account of ecological self was offered by J. Baird Callicott, who criticized Naess on several points (2017), such as the eclectic sources Naess drew upon or the fact that he ignored the cutting-edge science of ecology to inform the ecological self. He proposed, instead, the concept of the ecological self that explicitly recalls the tradition of the Kyoto School of Japanese Buddhism in conjunction with the potential implications of current ecological knowledge. Callicott suggested an *ecological* notion of the self "as a knot, nexus, or node in a skein of social and environmental relationships" [23] (p. 235). Since these relationships are internal (the self), to undo them would mean to undo the self, and nothing left would remain (he used the Buddhist expression *topos* of *mu*, place of nothing). Using his own words:

> "[. . .] the ecological self is constituted by its internal socio-environmental relations. Untie the knot that is oneself in the socio-biospherical net or field of internal relations, and there's nothing left of the self" [23] (p. 241).

More recently, Callicott suggested that it is possible to find conceptual foundations for such an ecological self in existing notions elaborated within both natural sciences and western scholarly traditions [6].

A third perspective on the notion of ecological self is that proposed by Christian Diehm. After analyzing Naess's work on the ecological self, he suggested a different idea of identification with others (including non-human others). According to Diehm, the "process of identification"—central in Naess' ecosophy because it makes possible the very development of the ecological self—should be understood as "a response, just one mode of an ongoing dialogue in which we attempt to find ways to articulate ourselves properly to others, a way of recognizing and assuming responsibility, of being responsive" [24] (p. 34).

A fourth and final option for the notion of ecological self can be found within ecofeminist scholarship. For instance, in her book *Feminism and the Mastery of Nature*, Plumwood wrote about the ecological self as a relational self, essentially characterized by non-instrumental relationships to others. These relationships are not part of the self since they are "incorporated" or "assimilated" within it nor because their flourishing contributes to the well-being of the self. In contrast, for Plumwood, ecological selves represent independent centers of intentionality and agency, which impose limits on the self, thus constituting it. The "earth others" [1] are in constant dialogue with each other, every one of them with its own center. These ecological selves exist in and from this dialogue, made of recognition and awareness (of others and differences)[6]. She wrote:

> "The ecological self can be viewed as a type of relational self, one which includes the goal of the flourishing of earth others and the earth community among its own primary ends, and hence respects or cares for these others for their own sake" [1].

In the context of environmental philosophy, Naess, Callicott, Diehm, and Plumwood offered four main conceptions of the ecological self. In the next section, we illustrate our position in comparison to these approaches.

### *3.2. Our Position on the Ecological Self*

Despite some differences, these positions share a number of aspects. First, the previous conceptions of the ecological self assumed a subjective identity, implying that they tend to offer a polarized notion that oscillates between the same and the other. For example, Naess

started from the modern ego and social self separated from nature and thus extended it to a metaphysical ecological self, focusing on the identity of the ecological *human* self, with ecological relationships. Adopting another strategy, Plumwood started from a critique of the hegemonic "master perspective" [25] (p. 99) and suggested that the otherness of and within ecological relations facilitates entering into a relationship. Second, all the positions presented above consider the ecological self as something peculiar to human beings, thus assuming an anthropocentric conceptualization of the self.

A notable exception to this trend can be found in Freya Mathews' *Ecological Self.* In her work, the author adopted a non-anthropocentric conception of the self that can be applied to organisms but also to the cosmos, and to some extent to ecosystems:

> "The paradigm instance of the self-realizing system—or 'self'—is the organism. But the geometrodynamic universe as a whole also qualifies for selfhood. A self-realizing being is one which, by its very activity, defines and embodies a value (viz., its value-for-itself.) Since self-realization is a function of ecological interconnectedness, the property of intrinsic value is likewise a function of such connectedness." [20] (p. 101).

Mathews inferred from that premise that "The individual is thus in a very real sense a microcosm of the wider self in which it occurs" [20] (p. 101), and that would imply an egalitarianism regarding the "intrinsic value" of substances in such a complex systemic metaphysical cosmic order. This normative consequence seems at first converge with "a bio- or eco-centric ethic" [20] (p. 103) that would have, as a principle, to "'tread lightly' on this earth, taking from it only what we must satisfy our 'vital needs'" [20] (p. 103). However, Mathews later emphasized the spiritual dimension of this "ethics of care" [20] (p. 105) and came back to describe human virtues of "awareness" and "love":

> "Meaningfulness is to be found in our spiritual capacity to keep the ecocosm on course, by teaching our hearts to practise affirmation, and by awakening our faculty of active, outreaching, world-directed love. Though a tendency to 'tread lightly' on the earth, and to take practical steps to safeguard the particular manifestations of Nature, will inevitably flow from such an attitude, the crucial contribution will be the attitude itself, a contribution of the heart and spirit." [20] (p. 113).

In other texts, she speaks about virtues of "commitment" and "loyalty" toward the earth community [26], which mean recognizing or being aligned with the conativity of the systems, working with it in a mutualistic and relational way, and promoting "grace" as an embodiment of the principle of "least resistance" [27] (p. 22)—inspired by the Taoist virtue known as *wuwei*.

Building on Mathews's conception, we propose a conceptualization of the ecological self that underscores the intrinsic connections and the various relationships in which all beings are meshed. In particular, we stress that both humans and non-humans can be/embody ecological selves. This relies on a relational ontology that seems common to Naess, Plumwood, Callicott, and Mathews [6] (p. 24). Regardless of distinct strategies to promote practical ways for humans to be aware of the ecological self (e.g., Naess' expansion or Plumwood's recognition of our inner relation to otherness), at this ontological level, the ecological self constitutes a fundamental premise of the self itself.

This conception of the ecological self can find a scientific ally in biologist and neurologist Francisco Varela's conception of organisms. Indeed, according to Varela, an organism is "a multiplicity of regional selves, all of them having some mode of self-constitution, and in their overall assemblage giving rise to an organism" [28] (p. 80). Varela described different regions of a "self" or "selves" that could also be useful in conceptualizing our version of "ecological selves". These are: "(1) a minimal or cellular unity, (2) a bodily self in its immunological foundations, (3) a cognitive perceptuo-motor self associated to animal behavior, (4) a socio-linguistic 'I' of subjectivity, and (5) the collective social multi-individual totality". Considering the functioning of such systems, we do not need to assert the autonoetic consciousness (region 4) as a necessary condition for selfhood. The forms

of autonomy and self-constitution that give rise to an organism (especially in regions 1 to 3 characterized by an autopoietic organization) allow us to speak of a mesh of relations among "selfless selves" [28] (p. 80), hence, other-than-human—"ego"—subjective selves.

Therefore, our conception of the ecological self includes, but is not limited to, the human self. We assume that the ecological self can be expressed both by humans and some non-humans at different levels (organisms, ecosystems, and even cosmos). Ecological selves could be identified in a symbiotic relationship acting within a web/field of ecological interdependencies in a process of self-realization (which may imply key dimensions such as conativity, agency, identity, and some forms of intelligence). Our ecological self aims to decenter and relocate the notion of selfhood in a less anthropocentric way. This does not mean that we deny differences between species and individuals. Rather, we stress the fact that some non-human beings can also express an ecological self, living in a web of relationships, in which they actively pursue their self-realization and interact accordingly. The processes of communication (release information, encode information, decode information) shared amongst living beings could be the source of multiple examples of such ecological self expressions. For instance, some birds may behave unusually when a storm or an earthquake is coming; some octopi are capable of constantly mimicking their surroundings and have developed very refined hunting strategies; bacteria develop resistance to antibiotics; and some trees like acacias may alter their composition when eaten and release different chemicals to communicate this information to other members of their species [29]. Our perspective on the ecological self highlights a connection rather than a disconnection between humans and other-than-human beings, recognizing in the human self something that is also expressed in other beings but in other forms and maybe pursuing other ends. Why and how can these ontological considerations about the ecological self affect our understanding of environmental virtue ethics?

## 4. Towards Functional Ecological Virtues

The aim of this paper is to challenge the modern western hegemonic tendency to interpret the anthropological difference—sometimes expressed in terms of "virtue" and *a fortiori* of "moral virtue"—as a criterion of human supremacy and uniqueness. We claim that it is possible to recognize an ontological analogy between humans and non-humans through the above-mentioned conception of the ecological self (e.g., a self that is expressed by both humans and non-humans caught in their web of interrelations). We call these "mastering skills" *functional ecological virtues*, i.e., a-moral virtues that can be exercised both by humans and non-humans. This perspective tries to consider, in a new way, the continuity and discontinuity between humans and non-humans, not as sharing an inanimate physicality or materiality, but as sharing agentivity and, in some cases, ecological virtues (excellences while being an ecological self). Hence, we could say that we tend to emphasize the *analogical*[7] potentialities of virtue ethics rather than its human *supremacist* tendency. However, does it make sense to speak of functional ecological *virtue* to describe certain excellences of these ecological selves? If so, what would it mean to consider non-human agents as potentially expressing (functional) ecological virtues, and how would this transformative[8] conception of ecological virtue influence human behaviors, especially facing planetary ecological crises?

### 4.1. Step by Step: What Do We Mean by (Functional) Ecological Virtue?

In Section 3, we explained that both humans and non-humans could be considered ecological selves. In their interconnections and relationships to others, plants, animals, and other living beings actively pursue self-realization. If we consider classical moral virtue—strongly or weakly anthropocentric, expressed by a (human) moral agent and pursuing a moral end—it seems that it would be a category error to try to apply virtue language to non-human ecological selves. We respond to some objections later, but what if we start by reconsidering virtue in a different way, namely as excellent behaviors and actions at a *functional level*?

The first strategy is to consider functional ecological virtues *in relation to an ecological good,* the assessment of which depends on scientific criteria, which are often contested and dependent on the theory, the methodology, the scale considered, the system studied, etc. Actions that contribute to a contextual ecological good or the flourishing of a local ecosystem could be qualified as "ecological virtues" in a non-moral (a-moral) sense. For instance, drawing from Leopold's *Land Ethics* and considering the ecocentric duty to consider and respect other-than human "citizens", Bill Shaw proposed to characterize "land virtues": "The attitudes and practices that serve the ultimate good in this new paradigm—land virtues—tend to preserve the integrity, stability, and beauty of natural systems. Vices tend to destabilize and to destroy these characteristics of natural systems" [30][9]. Fundamentally, this strategy is not satisfactory because either it supposes a heavy set of metaphysical assumptions to consider that evolution is inherently and morally good, but this position is obviously very contested (cf. [31] for instance). Or, it must confront itself with blurred notions of normative criteria to characterize what is *ecologically* good (biodiversity, ecological health, resilience, integrity, connectivity, etc.).

The second strategy could be to characterize *functional ecological virtues* as *expressed by an ecological self.* They might be characterized as excellence in interacting with the environmental context, flourishing and self-realizing within a mesh of interdependencies. For instance, Michael Marder spoke about the "wisdom of plants" [32] and showed that plants are particularly excellent in "living-with" [32] (p. 51) the elements and threats constituting their middle of life. In this sense, plants express a specific virtue in seeking and soaking water and minerals in their surroundings, amongst other excellences [33]. These abilities can be considered *virtuous* because they can develop and perfect themselves according to specific vital ends, or they can fail and lead to some vital failure. On this account, functional ecological virtues are not necessarily (new) specific human qualities or attitudes that we (humans) need to develop in order to face the practical inertia or the non-reaction, which seems common among people in front of the ongoing ecological crisis [34].

Though not restricted to human agents, functional ecological virtues are (ecological) *"agent-focused"* [11] in the sense that they express the ecological self's excellence in specific ecological contexts. For example, a domestic dog named *Gaïa* can be regarded as capable of caring within a familial ecosystem[10]. Moreover, these virtues are "*target-based*", in the sense that certain actions and behaviors can be considered virtuous in that they succeed "in responding well to the demands of the world" [35]. Examples of this are quinoa, which can resist drought, very high salinity, and poor soil [36], or Burmese Pythons, which adapt so well to the anthropogenic warming of Florida Everglades' ecosystem that they become a threat to some native species, such as medium-sized mammals [37]). If we try to transpose Swanton's pluralistic categories to virtue [13], four of them might make sense applied to functional ecological virtues, which can be:

1. "*Value-based*". When ecological selves value and enhance at least some *vital values* (e.g., engagement and caring relationships between emperor penguin's parents and their chicks express some valuation of continuing life, or the Vogelkop bowerbird (*Amblyornis inornata*) in West Papua that builds a hut and decorates it to convince reproductive female values the creation and organization of an adjusted habitat).

2. "*Bond-based*". When ecological virtues express a fine attunement to the mesh of interdependencies of the world (e.g., macaroni penguin faithful couples that reunite about 3 months each year to reproduce, give birth, and raise their chicks before living on their own the rest of the year express a notable virtuosity in forging lasting ties; or mycorrhizal symbiosis, for example, between oak trees and truffles, express a very refined and fructuous biochemical and molecular dialog that co-benefit the individuals and species involved, the soil, the forest, etc.).

3. "*Flourishing-based*". When non-humans seem able to act *for the good* of others (e.g., any cooperative action, such as feeding techniques or common hunt by a wolf pack or a group of humpback whales chasing krill to the surface with bubbles of air, that demonstrates

a sense of collectiveness within a specific group; symbiotic relationships, as mentioned before, would demonstrate co-flourishing virtue).

4. "*Status-based*". When the recognition of social or hierarchical relationships, expressed in the non-human world, activates the ability to perform accordingly and to play with them (e.g., understanding of territorial signatures, respect, or contestation of the pack organization seem to be current occasions in wolf lives that express such virtues; the relationship between the beehive and its queen may express some virtues of protection, reproduction, or unification of the hive).

Of course, not all non-humans can exercise ecological virtues, and not every action performed well may correspond to a functional ecological virtue. Our proposal is just an exploration of a rarely navigated field of research (e.g., can other-than-human beings act virtuously?). What we aim to show is that, depending on how we define (ecological) virtues, there are potentially a lot of examples of non-human beings that can exercise and master ecological virtues. Although recognizing that non-humans can "master ecological virtues" (at least in some specific contexts[11]) does not imply a direct normative judgement (i.e., it does not prescribe anything), it does contribute to thinking about the human functional and moral sphere differently.

### 4.2. Functional Ecological Virtues and Moral Ecological Virtues

Why is it important to specify that such ecological virtues are *functional*? Classically, virtue is defined in relation to a *function* (*ergon* in Aristoteles' philosophy). Therefore, it could seem redundant to qualify ecological virtue as *functional*, but we do so explicitly in order to clarify and at the same time challenge the traditional (in VE and EVE) identification of virtue with "moral virtue". Indeed, opening up the notion of virtue may question and disrupt the anthropocentric, dualistic, and naturalistic ontology that has become predominant in VE debates. We would like to explore the possibility of conceiving, in the first step, *a-moral virtues*, meaning virtues that are not yet considered from any moral perspective and virtues that qualify excellence defined in reference to a functional end (which could end up being morally good or bad). It does not mean that these virtues are necessarily anti-moral (like it is commonly understood when virtue ethicists discuss non-moral virtues, e.g., the excellence in killing furtively for a hitman). Functional ecological virtues are considered before any definition of any moral good and bad in order to (1) bring more complexity and nuances in our spontaneous understanding of what is ecologically good or bad and also to (2) contest and deconstruct rooted assumptions. Using the concept of virtue as a decentering tool, we question the dualistic naturalistic anthropocentrism in western cultures and suggest that virtuous dispositions and behaviors might be more shared or distributed amongst living beings than previously thought or admitted.

How do functional ecological virtues enter the *human* moral sphere? By stressing "human", we do not wish to exacerbate any separation between non-humans and humans. We just want to investigate the implication of our proposal (functional ecological virtues) within a virtue ethical theory, that, by definition, is for humans. In other words, what are the moral implications (for humans) of this new kind of virtue? Adopting functional ecological virtues could have important consequences:

(i) The transformative function of these ecological virtues for humans. The acknowledgment of an ontological closeness to other beings (because of this shared ecological self) and how non-human beings can master this condition of interrelations with others can originate or stimulate in humans interest, sensitivity, care, attention, etc., towards non-humans, towards "the other-than-humans". The recognition of functional ecological virtues would eventually reinforce moral ecological virtues in humans.

(ii) Another possible moral return of functional ecological virtues for humans is the educational aspect. Recognizing "virtues" in non-humans means recognizing complexity and excellences beyond the human world, and at the same time, it means to stress the human participation to a shared ecological world animated by the active self-realization of multiple other ecological selves. Once again, this could have an influence on how humans

relate to non-humans and could reinforce moral ecological virtues in children as well as in adults.

(iii) A third moral implication of this reconfiguration of the field of virtues could be to reconnect naturalistic modern cultures with other non-modern cultures, while recognizing a potentially common ground to compose a common world, contesting the arrogant modern presupposition that the progress of knowledge and civilization is to de-animate the non-human world, and considering that all non-humans only react to a mechanistic determinism. It could then open the way for "partnerships ethics", whose principles, as stressed by Carolyn Merchant [39–41], can rule relationships with non-human agents as well as with other cultures.

*4.3. Possible Objections*

We are aware that the thesis proposed in this paper can raise more than one objection. In the following section, we address some of these potential criticisms.

(1)     The application of "virtue language" to describe the behaviors and attitudes of non-human beings is counterintuitive and potentially wrong. Other-than-human beings or more-than-human beings have been classically identified as non-moral agents or, at most, as moral patients. Indeed, the status of moral agency in modern western cultures is attributed to human *only*, insofar as humans can allegedly conduct their own behavior according to autonomous norms and practical reasoning [42], while virtues designate acquired excellences and not endowments. Thus, although one might agree with recognizing ecological selves in non-human behaviors, it still remains unclear how it would be possible to speak about virtues for them; whether they are considered as ecological selves or not, they would still fall under the category of moral patients, potentially protected or cared for by the expression of environmental virtues (exercised by human agents) but not as potential virtuous agents. However, as we have stressed, we are not talking about moral ecological virtues for non-humans. We rather suggest an extension of the space and meaning of ecological virtues on the basis of ontological considerations about the self. Functional ecological virtues are not virtues in any traditional moral sense; they are a-moral virtues that express mastering skills of ecological selves that can affect either humans, non-humans, or both (see Section 2 above).

(2)     A second objection might be put as a question: why use (and possibly distort) the concept of virtue instead of drawing upon another concept to describe such excellences or mastering skills? As we mentioned above, excellence and mastering skills are already historically part of the concept of virtue. These different concepts are not mutually exclusive but embedded in a mutual understanding in the history of ideas. The intellectual challenge of this paper was to question provocatively a well-accepted assumption of environmental virtue ethics (e.g., that the language of virtues is limited to human beings) and see what this could bring about. Using a usually (anthropocentric) moral term in an a-moral way is a strategy to expand and deconstruct dualistic inherited and rarely questioned structures or engrained theoretical assumptions. Moreover, if it were possible to consider a-moral excellences in terms of virtues, this may have a reinforcing positive impact on human moral ecological virtues, in the way explained in (i), (ii), and (iii) (see previous Section 4.2 on pp. 10–11).

(3)     One might say that a virtue, by definition, assumes the intentionality of the (human) moral agent as a prerequisite. This is, for example, the objection proposed by Holmes. He contested the extension of virtues to non-human beings, arguing that virtues are "achievements not endowments", "acquired excellences" not "genetics endowments" [43] (p. 69). Holmes preferred to speak about "*values* as intrinsic achievements in wild nature" (ibid.) rather than using the concept of animal virtue. We can note that he did not consider plants, bacteria, or other living beings. Holmes feared that environmental virtue ethic approaches may not suffice to value nature in itself apart from human interests. Although he mentioned the meaning of virtue used

by Thoreau "in the archaic sense of an "excellence", survival skills in the migratory fish (with no reference to praiseworthy character achievement thus analogous to perfect pitch[12] in humans)", he contested it and asserted that value-based ethics respecting the intrinsic value of nature are preferable rather than virtue-based-ethics. We can reply to the objection that Holmes remains dualistic and anthropocentric in a classical naturalistic (as understood by Descola) way. The ethological dualism between will/instinct that underpins the dualistic opposition of acquired excellences/endowments is widely contested. More and more ethologists/psychologists/cognitivists tend to pay attention to individual variations and developed abilities in animal behaviors [44–47]. We argue that human beings, animals, plants, and other living beings can be characterized by "developed excellences" and that we should also pay attention to individual variations rather than only consider species-typical behavior. Moreover, Holmes seemed to adopt agent-based virtue ethics as a strawman, saying that concerns for human ends are not enough to cope with ecological issues. However, as Christine Swanton suggested,

"The environmental virtues can be understood as being virtues not just because they are dispositions to promote human-centered ends, but also the ends of the flourishing and integrity of ecosystems, species, and natural objects (sentient and non-sentient) for their own sakes. Furthermore, [the principle of pluralistic virtue] allows for the environmental virtues to have a complex profile, consisting not just of promotion of good or value, but also of respect, love for, and appreciation of natural objects." [13] (p. 94).

We therefore suggest that functional ecological virtues do not need intentionality as a requisite, although there might be examples of nonhuman beings intentionally performing actions that could be defined as virtuous (e.g., from human-trained rescue dogs to animals spontaneously rescuing humans and other animals without being trained to/rewarded by humans)[13].

## 5. Conclusions

This paper aimed to reweave the dualistic gap that separates humans from nonhumans in the moral sphere or, at least, in environmental virtue ethics debates. It also aimed to contest the modern dualistic human supremacism constructed on an ontological structure, as shown by Descola, which considered that the continuity between human and non-human beings is based on (inanimate) materiality and the fact that only human beings have an interiority (hence a moral sphere)

We propose to expand or extend the concept of virtues, loading it with a functional a-moral meaning. In so doing, we see a possible way to include, among virtues, excellent actions, behaviors, and attitudes actively exercised by non-humans (plants, non-human animals, even the entire cosmos), expressing their ecological selves. The presupposition of this theoretical move is that we may be able to recognize an ontological analogy between humans and non-humans through an understanding of the ecological self that can be expressed by both humans and non-humans caught in their web of interrelations. Therefore, our strategy is to relocate virtue as an inner ensemble of a more comprehensive way of developing excellence in a mesh of various developed excellences.

Extending the ecological self to non-humans implies, first, that mastering "skills", actions, or behaviors of living in such interconnected webs with a multitude of other beings and pursuing their own interests (flourishing themselves and sometimes making others flourish) can be recognized as functional a-moral virtues that can be exercised both by humans and non-humans. Second, if we operate this transformation, then we might also transform, in return, the concept of human virtue, not as a supremacist acquired excellence, but as a development of the contextual excellence to interact with the middle of life. Thirdly, virtue entering the moral sphere at a reflexive level could be one peculiar trait of human collectives' contexts. Fourth, the awareness of "living-with" others, and the recognition of others themselves as ecological selves, can promote human behaviors and

attitudes that positively affect the various and pressing environmental crises, especially while contesting the reduction of non-humans to a stock of resources to be exploited by humans. This proposal might also have implications for environmental justice, especially while helping to reconsider Indigenous or non-modern knowledge and cosmovisions. This paper contributes to environmental ethics scholarship in several ways. It questions long-standing ontological assumptions about agency and the type of virtues different agents can exercise (linked to Section 2). It broadens the notion of agency as ecological self, thus enlarging the boundaries of who can act virtuously (linked to Section 3). It decenters human agency—assuming that non-human beings could exercise excellent dispositions as ecological virtues—making it possible to conceive non-anthropocentric a-moral dispositions as functional ecological virtues (linked to Section 4). In the case of human ecological selves acting out functional ecological virtues, *intentionality* can be a relevant feature of such virtuous behavior. However, in the case of non-human ecological selves, talking about intentionality might not be reasonable; therefore, it does not constitute a feature.

Going back to the initial question of this SI—"Is Environmental Virtue Ethics a Virtuous Anthropocentrism?"—our response in this essay was "It Depends". It depends on what kind of ontological premises we are moving from. By offering an "ontological detour" of some of the theoretical premises of EVE, we suggested that excellent behaviors as functional ecological virtues can be exercised either by human agents or by some non-human beings. Both would, indeed, act virtuously as ecological selves.

**Author Contributions:** Conceptualization, writing—review and editing, N.C., D.D. and G.F. All authors have read and agreed to the published version of the manuscript.

**Funding:** This research received no external funding.

**Institutional Review Board Statement:** Not applicable.

**Informed Consent Statement:** Not applicable.

**Data Availability Statement:** Data are contained within the article.

**Acknowledgments:** The authors would like to thank Nora Ward of the University of Galway for proofreading the article.

**Conflicts of Interest:** The authors declare no conflicts of interest.

## Notes

1.   "Non-human" is intended here as a synonym of "other-than-human" in the sense of a useful logical distinction but not in the sense of maintaining or defending a dualistic stance.

2.   In this text, we do not distinguish between "environmental virtue(s)" and "ecological virtue(s)". In the reminder, we use only the term "ecological virtue(s)" assuming the other as a synonym. However, in the paper entitled *Virtue Ethics and Ecological Self: From Environmental to Ecological Virtues* in this collection, Gérald Hess proposed a distinction precisely between these two. Cfr. [2].

3.   See this note of Roger Crisp in his translation of *Nichomachean Ethics*: "Aret*é* = virtue. Alternative translation: 'excellence'. Covers non-moral as well as moral characteristics, as in, e.g., 'This book has many virtues.' Aristotle usually has in mind either moral excellences of character or intellectual excellences when using the term. It is related to the notion of 'characteristic activity' (ergon): the virtue of something consists in its capacity to perform well its characteristic activity (the virtue of an eye, for example, is to see well). Analogously, a vice (*kakia*) may be seen as a defect or flaw" [14] (p. 205).

4.   If "non-humans" is considered a synonym of "nature", then option [d] could be understood within [d].

5.   "I therefore tentatively introduce, perhaps for the first time ever, a concept of *ecological self*" [21] (p. 35).

6.   Maybe the most important debate about the ecological self is the deep ecology–ecofeminism discussion of the 1980s–1990s, when ecofeminists criticized some aspects of the ecological self proposed by deep ecologists [24]. The main critique dealt with/focused on the fact that an "expanded self" or an "indistinguishable self", and even the identification proposed by Naess, maintains and reproduces some patriarchal distortions.

7.   Analogism refers here to Descola's work [3] which gives it an ontological (and not only logical) meaning. Descola described the modern western ontology, called *naturalism*, as a reversed formula of *animism*: "articulating a discontinuity of interiorities and a continuity of physicalities" [3] (p. 172). Another ontology is *analogism,* which was dominant in western ontology until the Renaissance and which is common in some asian traditional cultures, among others. It is characterized as "a mode of identification that divides up the whole collection of existing beings into a multiplicity of essences, forms, and substances separated by small

distinctions and sometimes arranged on a graduated scale so that it becomes possible to recompose the system of initial contrasts into a dense network of analogies that link together the intrinsic properties of the entities that are distinguished in it" [3] (p. 201). In this sense, *ecological virtues* could be considered as such an analogical "form" that can be shared and recognized amongst different beings and potentially arranged in another scale than the naturalistic dualistic hierarchy that only conceives human supremacy over other natural beings.

8   By 'transformative' we mean mainly two things: (1) that we transform the mainstream conception of environmental virtue, and (2) that this new conception of ecological virtue presupposes a transformation of our main naturalistic (in Descola's sense) ontological structures and corresponding dualistic experiences of the self.

9   When Shaw exemplified these land virtues, with "respect (ecological sensitivity), prudence and practical judgment", he tended to consider only human actions, allowing us to understand that land virtues are only characterized within the biotic community and human citizens and not all the other ones!

10   Our proposition could even be applied to a broader scale, such as Lovelock's Gaïa self-regulating her living conditions to provide a habitat for biodiversity.

11   For instance, at a functional level, some Ruppell's griffon vultures (*Gyps rueppelli*) that can fly up to more than 10,000 meters above African lands demonstrate incredible skills in evolving in the aerial fluid element and can undoubtedly be seen a master by paraglider's pilots. Another more politically involved example are 'weeds' and the rhizomatic vegetal resistance and proliferation model, which are taken by alternative minority ecological collectives (like in the french ZAD of Notre-Dame-des-Landes) as explicit counter-hegemonic inspiring beings and dynamics for eco-social emancipatory movements (see for instance [38] entitled "Eloge des mauvaises herbes", *In praise of weeds*).

12   Can functional ecological virtues be considered endowed skills analogous to "perfect pitch" and thus not praiseworthy acquired excellences that seem to define virtues? This example may not be the best choice. First of all, we can observe that nobody will ever develop a so-called "perfect pitch", which is a very contextual and culturally situated skill, outside of a specific musical, familial, and social background. Thus, "perfect pitch" seems a weak paradigm of endowments as opposed to acquired excellence. It is rather the actualization in certain individuals of potentialities by a specific context (of learning, practicing, and playing some music).

13   We also note that intentionality may not be a requisite in classical virtue ethics, at least in the exercising of virtue, because virtues tend to be considered as internalized disposition or *hexis* in aristotelian terms. Moreover, some virtue ethics (e.g., daoist, zen buddhist, etc.) can consider non-deliberative virtues, like *wuwei* or *ziran*, or spontaneity as a key virtue.

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
