# Peer review of "Ecological Virtuous Selves: Towards a Non-Anthropocentric Environmental Virtue Ethic?"

_philosophies, doi:10.3390/philosophies9010011_

Round 1
Reviewer 1 Report
Comments and Suggestions for Authors
The quality of the English is very good, but there remain occasional grammatical and stylistic glitches and quirks that require correction by a native Anglophone. The level of the discussion is very abstract which would make the argument difficult to follow for readers who are not intimately familiar with environmental virtue ethics and varieties of an "ecological self." If the editors think it appropriate the author could update their reference to my version of the ecological self by discussing and citing J. Baird Callicott, The topos of mu and the predicative self," Dialogue and Universalism: Journal of the International Society for Universal Dialogue 33/2 (2023): 9-35. I am aware that this would reveal my identity to the authors, but I am comfortable with that.
Comments on the Quality of English LanguageVery good, but not perfect. Need editing by a native Anglophone
Reviewer 2 Report
Comments and Suggestions for Authors
The article is a worthy summary and analysis of a non-Anthropocentric Environmental Virtue Ethic. It is impressively researched, with a great breadth and depth of sources cited
The article is a good introductory offering for the intended audience as per the journal title of ‘philosphies.’ Where this for a different journal, e.g. one on environmental values or environmental ethics, I would recommend a far more rigorously argued and less ‘shopping list’ like article. However, I sense that the authors have deliberately fashioned their material for this journal’s readership, and believe that this is one of the strengths of the article.
If the above, very minor, recommendations can be addressed then this will make a very fine article for the journal.
Minor grammatical, spelling and typos throughout e.g.
“in the sense that they may favor of a human centered conception” should be “in the sense that they may favor a human centered conception”
Spelling e.g. “misantropic”
“From at least the latest 1980s onwards…”
This claim is wholly unsubstantiated: “This may serve as the ground on 107 which to build non-anthropocentric accounts of EVE that imagines and devises environ- 108 mental and climate policies differently.” – Earth is currenty in a no-analogue state. And undergoing the sixth mass exintcion event. Where is the engagement with these issues? The climate crisis (aka “global boiling) is absenst from discussion, except where claims of relevance, such as the above, are made, but then not delivered on.
Also, the excessive use of acronyms and numbered lists reduced the readibility of the article. However, as per above ‘shopping list’ comment, this could a deliberate attempt for the authors to get a wider readership beyond environmental values or environmental ethics.
If the following can be attended to, then the article would make a worthy contribution to the volume.
Comments on the Quality of English Language
See above
